# Quantitative Characterization of the Anisotropy of the Stress-Optical Properties of Polyethylene Terephthalate Films Based on the Photoelastic Method

**DOI:** 10.3390/polym14163257

**Published:** 2022-08-10

**Authors:** Quanyan He, Miaojing Wang, Yitao Du, Qinghua Qin, Wei Qiu

**Affiliations:** 1Department of Mechanics, Tianjin University, Tianjin 300354, China; 2Tianjin Key Laboratory of Modern Engineering Mechanics, Tianjin 300354, China

**Keywords:** polyethylene terephthalate film, stress-optical properties, stress-induced birefringence, anisotropy, photoelasticity, flexible electronics

## Abstract

Polyethylene terephthalate (PET) is one of the most commonly used substrate materials in the field of flexible electronics, and its stress-induced birefringence often has a detrimental effect on the optical properties of the device. Therefore, a deep and systematic understanding of the stress-optical properties of PET films is crucial for device design and manufacture. The photoelastic method is a direct optical measurement technique based on the stress-induced birefringence effect of materials, which has the advantages of being nondestructive and noncontact. In this work, the photoelastic method was used to quantitatively characterize the anisotropy of the stress-optical properties of PET films under the uniaxial stress state. First, a self-built reflection-transmission coaxial bidirectional photoelasticity measurement system was developed by means of a combination of transmission and reflection photoelasticity. Then, the stress-optical coefficients and isoclinic angles of PET films with different stretching angles were measured. Finally, the linear combinations of the photoelastic tensor components and refractive-index-related parameters were determined by fitting the analytical relationship between the stress-optical coefficients and isoclinic angles.

## 1. Introduction

Flexible electronic devices have developed rapidly in recent years and are widely used in flexible displays, medical diagnostics, energy production and other fields [1], and they are bound to be a cutting-edge technology that will change human life in the future. Polyethylene terephthalate (PET) is one of the most commonly used polymers as a substrate material in the field of flexible electronic devices [2]. The mechanical and optical properties of PET are the key factors affecting the structure, performance and efficiency of flexible electronic devices. For example, its elastic modulus and strength are the main basis for structural design [3], and its refractive index and birefringence are essential a priori parameters in improving the performance and efficiency of the device [4,5]. Since flexible electronic devices are usually under tensile and bending loads, stress-induced birefringence usually has detrimental effects on optical properties. The birefringence of the flexible substrate in a flexible liquid crystal display (LCD) changes the polarization state of the emitted light [6,7]. Therefore, comprehension of the relationship between the PET film stress and optical properties provides accurate control of the optical properties of PET films during deformation, which is crucial for the design and manufacture of high-quality, low-power flexible electronic devices.

The stress/strain-optical coefficient describes the linear relationship between birefringence and stress/strain when the material is in the elastic stage. The main methods adopted for experimental studies that focus on stress-induced birefringence in polymer films are refraction-based methods [8,9,10], X-ray diffraction methods [11], spectral analysis [12,13], ellipsometric methods [14,15,16] and reflection difference spectroscopy [17]. Based on a dynamic birefringence instrument, Onogi et al. investigated the variation in the strain-optical coefficient with the frequency of periodic tensile strain [18]. Ryu et al. compared the relationship between stress and birefringence during uniaxial tensile testing of amorphous PET films at different tensile speeds in the temperature range of 80–95 °C [19]. Nou et al. measured the birefringence of PET films with white light and a spectrometer and analyzed the effect of wavelength on strain-induced birefringence of polymer in combination with spectroscopy [13]. Based on an Abbe refractometer and infrared spectrometer, Cail et al. demonstrated the stress-optical behavior of PET films [10]. Using reflection differential spectroscopy, Schmidegg et al. found that the strain-induced birefringence of PET films has a strong azimuthal dependence [17]. Zhou et al. studied the strain-optical behavior of PET films during uniaxial tension in different directions by using Müller matrix ellipsometry [20]. In summary, the common problem of most current methods is the inability to visually measure the birefringence and optic principal axis orientation of PET films and the lack of a complete quantitative characterization of the anisotropy of the stress-optical properties of PET films.

The photoelastic method has the potential to be an effective experimental means to quantitatively characterize the anisotropy of the stress-optical properties of polymer films such as PET. Photoelasticity was first described by David Brewster [21,22] in the early 19th century, and its theoretical model was developed by Neumann [23] and Pockels [24] based on the same phenomenological approach. This method is the earliest developed and applied photomechanics technique, and its principle is the stress-optics law. Specifically, based on the sensitivity of birefringence to stress in certain materials, the corresponding stress state is obtained by determining the isochromatic and isoclinic lines in the polarized light field, where the material stress-optical coefficients are known or have been calibrated. Conversely, if the stress state is known, the photoelastic method may be used to quantify the stress-optical coefficients of the material as well as its anisotropy. The photoelastic method also has the advantages of full-field, real-time, and nondestructive properties for the accurate identification of fast-axis directions [25,26,27]. In 1986, Hecker and Morche [28] first proposed the phase-shifted photoelastic method to capture intensity information of images for evaluating isoclinic and isochromatic parameters on the domain. The photoelastic method has been used to characterize the stress-optical properties of polymer films. Gao et al. measured the stress-optical coefficients of optical polyimide (PI) films by uniaxial stretching experiments [29]. In particular, Lee et al. calibrated the stress-optical coefficients of PET and PEN films in two orthogonal directions using the split-beam two-dimensional photoelastic technique, and the results showed that their stress-optical coefficients were different in the 0° and 90° directions [30]. As a direct optical measurement technique, the photoelastic method shows superior capability in the characterization of the stress-optical properties of polymeric films.

In this paper, aiming at the problem of stress-induced birefringence of flexible polymer substrates, a self-built reflection-transmission coaxial bidirectional photoelasticity measurement system was developed based on the principle of photoelasticity and the combination of transmission and reflection. Transmission and reflection photoelastic experiments on PET films with different stretching angles were carried out to determine their stress-optical coefficients and optic principal axis directions. Based on the theory of photoelasticity, the analytical equation between the stress-optical coefficient and the isoclinic angle of the PET film under uniaxial stress was established. The relevant parameters were determined by fitting the experimental data to quantitatively characterize the anisotropy of the stress-optical properties of PET films under uniaxial stress.

## 2. Materials and Methods

### 2.1. Specimen Preparation

The PET film material studied in this work was supplied by Shenzhen Lelin Technology Development Co., Ltd. with a size of 300 × 200 × 0.175 mm^3^. The glass transition temperature *T*_g_ of the material is 70 °C, and the elastic modulus is 3544 MPa. The long and short sides of this rectangular PET material sample were set as the *X* and *Y* axes, respectively, and the sample coordinate system *X*-*O*-*Y* was established. The specimens were cut into long strips with a length of 40 mm and a width of 7 mm, and the long side was stretched at an angle *φ* to the *X*-axis. In this work, *φ* is called the stretching angle, which was from −45° to 45° with an interval of 15°. It should be noted that the above elastic modulus was determined using specimens with *φ* = −45°.

### 2.2. Experimental Equipment

A self-built reflection-transmission coaxial bidirectional photoelasticity measurement system was used in this work, the schematic diagram of which is shown in Figure 1. A halogen lamp was selected as the light source for the system, with a maximum output power of 300 W. A collimated beam expander was used to adjust the size of the light field, and then the collimated light was passed through a 532 nm bandpass filter with a half bandwidth of 10 nm to form a near-monochromatic beam. A pluggable mirror was used to switch the measurement mode, which was pulled out for transmission measurements and plugged in for reflection measurements. In the transmitted light path, the beam passed through a polarizer, the first quarter-wave plate, a beam splitter, a specimen, the second quarter-wave plate and the first analyzer and finally reached the first CMOS camera. In the reflected light path, the beam passed through the polarizer, the first quarter-wave plate and the beam splitter, and after passing through the specimen, it was reflected by the mirror. The reflected beam passed through the specimen again and then passed through the third quarter-wave plate and the second analyzer and eventually reached the second CMOS camera. A 1/2″CMOS (Baumer VCXG-13NIR, 1024 × 1280 pixels) camera was used to capture the photoelastic images. The above was the beam path under the circular polariscope, while all the quarter-wave plates under the plane polariscope were removed. In addition, all optical components were applicable to a 532 nm wavelength, where the beam splitter was a depolarizing beam splitter (Wuhan Youguang Technology Co., Ltd., Wuhan, China) that split the incident light into a transmitted beam and a reflected beam in a ratio of 50:50 without changing the polarization state of the incident light.

To clearly describe the angle settings for the optical components based on the reflection-transmission coaxial bidirectional photoelasticity measurement system, it is necessary to define the following coordinate systems. The following three coordinate systems are available in this measurement system, the z-axis of which is related to the direction of light propagation. The first coordinate system, denoted as the *x_c_*-*y_c_*-*z_c_* coordinate system, is defined according to the common optical path propagation direction of the transmitted and reflected optical paths. The second coordinate system, denoted as the *x_t_*-*y_t_*-*z_t_* coordinate system, is defined according to the propagation direction of the transmitted light path after passing through the beam splitter. The third coordinate system, denoted as the *x_r_*-*y_r_*-*z_r_* coordinate system, is defined according to the propagation direction of the reflected light reflected by the mirror. Thus, the above three coordinate systems are named the common optical path coordinate system (CCS), the transmitted optical path coordinate system (TCS) and the reflected optical path coordinate system (RCS), respectively.

As shown in Figure 2, in the CCS, the angle of the polarization direction of the polarizer and the fast axis of the first quarter-wave plate relative to the *x_c_* axis are denoted by *α* and *ζ*, respectively. In the TCS, the angles of the fast axis of the second quarter-wave plate and the polarization direction of the first analyzer relative to the *x_t_* axis are denoted by *γ* and *β*, respectively. In the RCS, the angles of the fast axis of the third quarter-wave plate and the polarization direction of the second analyzer relative to the *x_r_* axis are denoted by *γ*′ and *β*′, respectively. Note that the angle of the slow axis of the specimen with respect to the *x_t_* axis is *θ* in the TCS, while the angle with respect to the *x_r_* axis is −*θ* in the RCS.

### 2.3. Tensile Test

Before each tensile test, both ends of the PET specimen were fixed on a uniaxial tensile loading device and adjusted to ensure that the long side of the specimen was parallel to the *x*_t_/*x*_r_ axis. The specimens were then stretched uniaxially at room temperature (23 °C), and optical image acquisition was completed for each of the ten step loads for PET specimens with different stretching angles. The data of 50 × 50 pixels in the central area of the specimen were selected to calculate the stress-optical coefficient (SOC) under the uniaxial stress state.

## 3. Results and Discussion

### 3.1. Model

Whether under uniaxial or biaxial loading, the PET film is in a plane stress state. It has been shown that the out-of-plane birefringence of PET films will not change due to in-plane loading [20]. Therefore, in this work, only the effect of stress on the in-plane birefringence of the PET film material is discussed. The birefringence of the PET film material is reflected by the fact that its fast and slow axes have different principal refractive indices, so there will be a certain degree of phase retardation between the light waves propagating along these two axes. In photoelasticity, full-field fringe patterns containing phase retardation (isochromatic lines) and optic principal axis directions (isoclinic lines) are captured as digital images and processed for quantitative evaluation [31]. The angle *θ* of the slow axis of the specimen relative to the *x_t_* axis is defined as the isoclinic angle, and the phase retardation *δ* is defined as the isochromatic phase.

In the experiment, the ten-step phase-shifting photoelastic method [32] was used to analyze the light intensity information of the whole field in the sampling area. Four groups of images were captured in the plane polariscope without a quarter-wave plate. The angles of the polarizer and the analyzer and the corresponding intensity equations are shown in Table 1. Six groups of images were captured in the circular polariscope, and the polarizer was set to π/2. The angles of the quarter-wave plate and the analyzer and the corresponding intensity equations are shown in Table 2. In the table, *α* is the polarization direction of the polarizer, *β* and *β*′ are the polarization directions of the first and second analyzers, and *ζ*, *γ* and *γ*′ are the fast axis directions of the first, second and third quarter-wave plates, respectively. In the light intensity equation, *I_a_* and *J_a_* are the light intensities accounting for the amplitude of light, *I_b_* and *J_b_* are the background light intensities, and *I_i_* and *J_i_*, (*i* = 1, 2, …, 10) are the light intensities under the corresponding image numbers.

The isoclinic angle and isochromatic phase were determined by the following steps. First, according to the light intensity equations listed in Table 1, the wrapped isoclinic angles of the transmission and reflection photoelastic experiments, i.e., *θ_t_* and *θ_r_*, can be determined by:(1)$$\theta_{t}=\frac{1}{4}\tan^{-1}\frac{I_{3}-I_{2}}{I_{4}-I_{1}}$$
(2)$$\theta_{r}=\frac{1}{4}\tan^{-1}\frac{J_{2}-J_{3}}{J_{4}-J_{1}}$$

The wrapped isoclinic angles obtained from Equations (1) and (2) by computer numerical calculations are in the range of −π/4 to π/4. Before calculating the isochromatic phase, the isoclinic angle must be unwrapped to the range of −π/2 to π/2. In this work, quality guided phase unwrapping algorithms [33] are used for the unwrapping process. Subsequently, according to the light intensity equation listed in Table 2, the isochromatic phases of the transmission and reflection photoelastic experiments, i.e., *δ_t_* and *δ_r_*, can be determined by:(3)$$\delta_{t}=\tan^{-1}\left(\frac{(I_{9}-I_{7})\sin2\theta_{t}^{\prime }+(I_{10}-I_{8})\cos2\theta_{t}^{\prime }}{I_{6}-I_{5}}\right),$$
(4)$$\delta_{r}=\frac{1}{2}\tan^{-1}\left(\frac{(J_{7}-J_{9})\sin2\theta_{r}^{\prime }+(J_{10}-J_{8})\cos2\theta_{r}^{\prime }}{J_{5}-J_{6}}\right),$$
where $\theta_{t}^{\prime }$ and $\theta_{r}^{\prime }$ are the unwrapped isoclinic angles. The isochromatic phase determined by the transmission photoelastic experiment is in the range of −π to π, while the isochromatic phase determined by the reflection photoelastic experiment is in the range of −π/2 to π/2. It can be seen that the unwrapping process of the isochromatic phase in the reflection photoelastic experiment is more laborious than that in the transmission photoelastic experiment.

The isochromatic phase of the PET film is caused by the original birefringence and stress-induced birefringence, where stress-induced birefringence can be described by the stress-optical properties. According to Pockels’ phenomenological theory of photoelasticity [34], the stress-optical coefficients can be derived from the following relationship between the impermeability tensor Δ***β*** and the stress tensor ***σ***:(5)Δβ=Pσ,
where ***P*** is the fourth-order photoelastic tensor or stress-optical coefficient tensor, which can be written as a 6 × 6 matrix using Voigt notation. In this work, it is considered that the PET film is homogeneous in the thickness direction and its photoelastic tensor is symmetric about the *X*-*O*-*Y* plane. The photoelastic tensor ***P*** can be used to understand the anisotropy of the stress-optical properties, which is expressed in matrix form in the principal axis coordinate system of the impermeability tensor as:(6)$$P=\left[\begin{array}{cccccc} p_{11} & p_{12} & p_{13} & 0 & 0 & p_{16}\\ p_{12} & p_{22} & p_{12} & 0 & 0 & p_{26}\\ p_{13} & p_{23} & p_{33} & 0 & 0 & p_{36}\\ 0 & 0 & 0 & p_{44} & p_{45} & 0\\ 0 & 0 & 0 & p_{54} & p_{55} & 0\\ p_{61} & p_{62} & p_{63} & 0 & 0 & p_{66} \end{array}\right]$$

The photoelastic tensor in a 6 × 6 matrix in the plane stress state can be simplified to a 3 × 3 matrix, and Equation (5) can be simplified as:(7)$$\left[\begin{array}{c} \Delta \beta_{1}\\ \Delta \beta_{2}\\ 0 \end{array}\right]=\left[\begin{array}{ccc} p_{11} & p_{12} & p_{16}\\ p_{12} & p_{22} & p_{26}\\ p_{61} & p_{62} & p_{66} \end{array}\right]\left[\begin{array}{c} \sigma_{x}\\ \sigma_{y}\\ \tau_{xy} \end{array}\right],$$
where *p*_11_, *p*_12_, *p*_22_, *p*_16_, *p*_26_, *p*_61_, *p*_62_ and *p*_66_ are the components of the photoelastic tensor, and *σ_x_*, *σ_y_* and *τ_xy_* are the stress components in the principal axis coordinate system of the impermeability tensor. Δ*β*_1_ and Δ*β*_2_ are the changes in impermeability caused by stress, and the principal axis of the impermeability tensor coincides with the principal axis of the refractive index ellipsoid. Therefore, in the uniaxial stress state, the isoclinic angle *θ* is equivalent to the angle between principal axis 2 of impermeability and the uniaxial stress direction, as shown in Figure 2. The stress component can be expressed as the uniaxial stress *σ*:(8)$$\left[\begin{array}{cc} \sigma_{x} & \tau_{yx}\\ \tau_{xy} & \sigma_{y} \end{array}\right]=\left[\begin{array}{cc} -\sin\theta & -\cos\theta \\ \cos\theta & -\sin\theta \end{array}\right]\left[\begin{array}{cc} \sigma & 0\\ 0 & 0 \end{array}\right]\left[\begin{array}{cc} -\sin\theta & \cos\theta \\ -\cos\theta & -\sin\theta \end{array}\right]=\sigma \left[\begin{array}{cc} \sin^{2}\theta & -\sin\theta \cos\theta \\ -\sin\theta \cos\theta & \cos^{2}\theta \end{array}\right]$$

According to Equations (7) and (8), we obtain
(9)$$\Delta \beta_{1}-\Delta \beta_{2}=\sigma [(p_{11}-p_{12})\sin^{2}\theta +(p_{12}-p_{22})\cos^{2}\theta -(p_{16}-p_{26})\sin\theta \cos\theta ]$$

The change in impermeability caused by stress can be related to the refractive index as follows:(10)$$\Delta \beta_{1}=\frac{1}{n_{i}^{2}}-\frac{1}{n_{1}^{2}},\Delta \beta_{2}=\frac{1}{n_{j}^{2}}-\frac{1}{n_{2}^{2}},$$
where *n*_1_ and *n*_2_ are the refractive indices of the fast and slow axes of the specimen when no stress is applied to the specimen, which is the optical anisotropy caused by the original birefringence of the PET film, and *n_i_* and *n_j_* are the refractive indices of the fast and slow axes of the specimen after stress is applied.

Since the change in impermeability is small, the change in refractive index difference can be expressed as:(11)$$\Delta n-\Delta n_{0}=(n_{i}-n_{j})-(n_{1}-n_{2})≅-\frac{n_{2}^{2}n_{1}^{2}}{n_{2}+n_{1}}(\Delta \beta_{1}-\Delta \beta_{2})$$

Based on the photoelasticity principle, the stress-optical coefficient *C*(*θ*) of the PET film can be expressed as:(12)$$C(\theta )=-\frac{n_{2}^{2}n_{1}^{2}}{n_{2}+n_{1}}[(p_{11}-p_{12})\sin^{2}\theta +(p_{12}-p_{22})\cos^{2}\theta -(p_{16}-p_{26})\sin\theta \cos\theta ]$$

Then, the relationship between the isochromatic phase *δ* and the uniaxial stress *σ* can be established as:(13)$$\Delta \delta =\delta -\delta_{0}=\frac{2\pi hC(\theta )}{\lambda }\sigma$$

Taking the partial derivative of *σ* on the left and right sides of Equation (13), we obtain:(14)$$\frac{\partial \Delta \delta }{\partial \sigma }=\frac{2\pi h}{\lambda }C(\theta ),$$
where *δ*_0_ is the phase retardation caused by the original birefringence, *h* is the thickness of the specimen, and *λ* is the wavelength of the incident light of the specimen.

**Figure 2 polymers-14-03257-f002:**
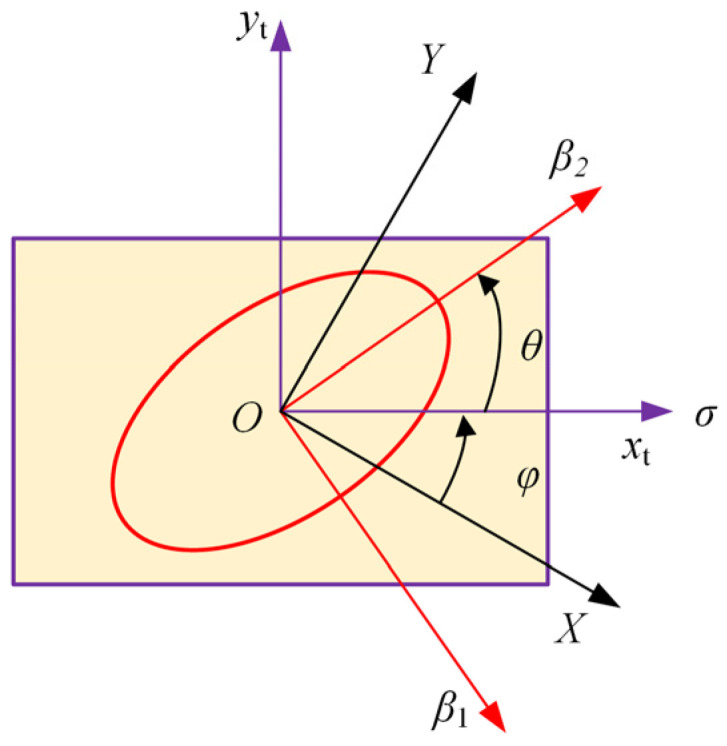
Definition of the principal axis of the permeability tensor and the direction of the uniaxial stress *σ* in the *X*-*O*-*Y* coordinate system. The directions of *β*_1_ and *β*_2_ coincide with the fast and slow axes, respectively. The direction of the uniaxial stress *σ* coincides with the *x*_t_ axis. *φ* and *θ* are the stretching angle and isoclinic angle, respectively.

### 3.2. Result

In this paper, both transmission and reflection photoelastic experiments were carried out to measure the isochromatic phase and isoclinic angle of PET films under uniaxial stress to quantitatively characterize the anisotropy of their stress-optical properties. First, the following explanation is required for all of the following isochromatic phase measurement results. As mentioned earlier, the quantitative relationship between the isochromatic phase and light intensity is established by an arctangent function. In the process of solving the arctangent function, the isochromatic phases of the transmission and reflection photoelastic experiments are restricted to the intervals [−π, π] and [−π/2, π/2], respectively, which is generally called “wrapped”. In the experiment, phase unwrapping is performed on the isochromatic phases measured at the last nine loads based on the isochromatic phases measured at the first load and their trends at the step load. Since the isochromatic phase of the PET film is caused by the original birefringence and stress-induced birefringence, the isochromatic phase measured at the first load has a large randomness. Therefore, all isochromatic phase measurement results cannot completely illustrate the birefringence of the specimen, but their trends with the stress are sufficient to give us an idea of the stress-optical properties of the PET film.

Figure 3a,b show the full-field isochromatic phase and isoclinic angle distributions obtained in the transmission photoelastic experiment for the PET film with a stretching angle of −15°, respectively. Similarly, Figure 4a,b show the full-field isochromatic phase and isoclinic angle distributions obtained in the reflection photoelastic experiment, respectively. In Figure 3 and Figure 4, it can be seen that both the isochromatic phase and isoclinic angle distribution images exhibit a grainy appearance, and the uniformity of the images obtained from the reflection experiment is slightly worse than that of the transmission experiment. The data of 50 × 50 pixels in the center region of the isochromatic phase distribution image were selected to calculate the stress-optical coefficients in the uniaxial stress state.

In this work, isochromatic phase and isoclinic angle distributions of PET films with various stretching angles were obtained in the transmission and reflection photoelastic experiments. Figure 5a,b depict the relative change in the isochromatic phase versus uniaxial stress for PET films at seven different stretching angles in the transmission and reflection photoelastic experiments, respectively. The experimental data for different stretching angles were linearly fitted with the goodness of fit R^2^ > 0.99, and the stress-optical coefficients were calculated from the slope of the fitted straight lines and Equation (14), as shown in Figure 5c,d. It can be seen that the PET films with different stretching angles have different stress-optical coefficients. The stress-optical coefficient of the PET film gradually changed from negative to positive as the stretching angle changed from −45° to 45°.

Figure 6 depicts the relation between the isoclinic angle and uniaxial stress of PET films with seven different stretching angles in the transmission and reflection photoelastic experiments. The dashes in Figure 6 are the average values of the isoclinic angle of PET films with different stretching angles. The result indicates that the increase in stress has little effect on the isoclinic angle for the same stretching angle of the PET film.

Figure 7 depicts the relation between the isoclinic angle versus the stretching angle for the PET film in the transmission and reflection photoelastic experiments. The result indicates that the isoclinic angle obtained from both the transmission and reflection photoelastic experiments decreases with increasing stretching angle. To describe the relationship between them more clearly, linear fitting was performed on the experimental data, with slopes of −0.9825 and −0.9997 and goodness of fit values of 0.9972 and 0.9993, respectively. According to the linear fitting equation between the isoclinic angle and the stretching angle, it is known that the stretching angles are −52.25° and −51.32° for the isoclinic angle of 0°, which correspond to the transmission and reflection photoelastic experimental results, respectively. This means that the slow axis direction coincides with the stretching direction for the PET film specimen with a stretching angle of −52.25° in the transmission photoelastic experiment and similarly for the reflection photoelastic experiment. The slope of the fit close to −1 indicates that the slow axis direction of the PET films with different stretching angles is almost unchanged in the *X*-*O*-*Y* coordinate system.

The isoclinic angle in the interval [−π/2, π/2] is converted to the interval [0, π] for the convenience of calculation. The relationship between the isoclinic angle and the stress-optical coefficient can be obtained based on the one-to-one correspondence between the isoclinic angle, the stress-optical coefficient and the stretching angle, as shown in Figure 8, which indicates that the stress-optical coefficient of the PET film depends on the isoclinic angle. A nonlinear curve fitting of these coefficients was performed according to the analytical Equation (12) for the stress-optical coefficients of the PET film using the Levenberg–Marquardt optimization algorithm. The best-fit results for the linear combinations of the photoelasticity tensor components and refractive-index-related parameters for the transmitted and reflected photoelasticity experiments are shown in Table 3, where the refractive-index-related parameters are in perfect agreement and the goodness of fit is greater than 0.99.

### 3.3. Discussion

The experimental results show that the birefringence of the PET film still maintains a linear response to uniaxial stress for different stretching directions in both transmission and reflection photoelastic experiments. Moreover, the stress-optical properties of PET films exhibit strong anisotropy. According to the definition of isochromatic phase, a decrease in isochromatic phase implies an increase in birefringence, and an increase in isochromatic phase implies a decrease in birefringence. When the stretching direction is close to the slow axis of the sample (−52.25°, −51.32°), such as stretching angles of −45° and −30°, the stress-optical coefficient is negative, and the birefringence increases because the uniaxial stress mainly makes the refractive index *n_j_* increase continuously. When the stretching direction is far from the slow axis of the sample (−52.25°, −51.32°), such as stretching angles of 45° and 30°, the stress-optical coefficient is positive, and the birefringence decreases because the uniaxial stress mainly makes the refractive index *n**_i_* increase continuously. Additionally, the consistency between the reflected and transmission photoelastic experimental results confirms the reliability of the quantitative characterization of the anisotropy of the stress-optical properties of PET films.

It must also be mentioned that even though the images obtained by the reflection photoelastic experiment are more affected by the particles in the polymer material, they show higher goodness of fit and smaller errors in both the fitting of the photoelastic tensor component and the recognition of the slow axis, and we attribute these advantages mainly to the fact that the light passes through the specimen twice. As mentioned in Section 3.1, the data processing (such as phase unwrapping) of reflection photoelastic experiments is more complicated than that of transmission photoelastic experiments. In contrast, the transmission photoelastic experiment has the advantages of a simpler experimental optical path and data analysis.

Some studies have reported the anisotropy of the stress-optical properties of PET films. Lee et al. [30] calibrated the stress-optical coefficients for the two orthogonal directions of PET films using the split-beam two-dimensional photoelastic technique, which were 2.616 × 10^−11^ Pa^−1^ and 1.456 × 10^−11^ Pa^−1^, respectively. Even though this result is on the same order of magnitude as the results in this work, both should not be considered positive. Zhou et al. [20] found that the stress-optical coefficient of PET films decreased from 4.321 × 10^−11^ Pa^−1^ to −5.041 × 10^−11^ Pa^−1^ as the stretching angle increased from 0 to 90° using Mueller matrix ellipsometry. This trend is almost the same as the results of this work, which can be mutually confirmed. Ellipsometry is a model-based indirect method that requires the modelling of optical properties, and then the information of interest can be obtained by solving inverse problems [35]. Different from ellipsometry, the photoelastic method is a direct method based on the principle of stress-induced birefringence. Thus, this method is less affected by other factors and it is easier to accurately quantify the stress-optical coefficient, and its experimental system is simpler, which is conducive to application in production. Due to the direct acquisition of the isoclinic angle by the photoelastic method, the present work is characterized by easy identification of the optic principal axis direction of the PET film and the smooth acquisition of the analytical relationship between the isoclinic angle and the stress-optical coefficient.

The stress-optical properties of polymer films influence the structural design and performance of flexible electronic devices. For example, in the design of flexible and stretchable electronic devices, the direction in which the stress-optical coefficient of the flexible film is close to zero should be taken as the stretching direction of the device as much as possible. In the manufacture of flexible LCDs, it is necessary to precisely grasp the anisotropy of the stress-optical properties of the flexible films used to guide the laying of each layer of the flexible films. Specifically, the basic structure of a flexible LCD is shown in Figure 9a, and the upper and lower flexible films are required as substrates in the fabrication of the liquid crystal cell. For the liquid crystal cell to switch properly between the dark and bright states, the fast axis directions of the two substrates must be perpendicular to each other [36], as shown in Figure 9b. However, the anisotropy of the stress-optical properties of flexible films has not been considered in relevant studies [37,38,39].

Based on the experimental results of the stress-optical coefficients of PET films already obtained in this work, the variation in transmittance with the angle between the fast axis direction of the lower substrate and the polarization direction of the lower polarizer was analyzed at different strains, as shown in Figure 9c. According to Jones matrix theory, the transmittance equation is expressed as:(15)$$T=\sin^{2}2\phi \sin^{2}(\delta_{0}+\Delta \delta_{s}),$$
where *ϕ* is the angle between the fast axis direction of the lower substrate and the polarization direction of the lower polarizer, abbreviated as the included angle. *δ*_0_ = *δ_U_ − δ_L_*, where *δ_U_* and *δ_L_* are the isochromatic phases of the upper and lower substrates due to the original birefringence, respectively. In this application example, without loss of generality, it is assumed that *δ*_0_ = π/4. Δ*δ_s_ = * Δ*δ_U_ −* Δ*δ_L_*, where Δ*δ_U_* and Δ*δ_L_* are the isochromatic phases of the upper and lower substrates due to the stress-induced birefringence, respectively. According to the analytical equation of the stress-optical coefficients obtained from the transmission photoelastic experiment, it is known that:(16)$$\Delta \delta_{s}=\frac{2\pi h\sigma }{\lambda }[1.1457\sin2\phi -8.262\cos2\phi )]$$

Due to the inevitable errors in the placement of flexible films, the transmittance should be less affected by the change in the included angle near the design angle. If the stress-optical properties of the flexible film are considered isotropic in the calculation of the transmittance, then Δ*δ_s_* = 0, which is equivalent to the transmittance when no strain/stress is applied. As shown by the black line in Figure 9c, the transmittance curve is symmetrically distributed at approximately *ϕ* = 45°, and the transmittance reaches a peak when *ϕ* = 45°. This means that the included angles are designed to be equivalent to 0° and 90°. As shown in Figure 9c, the peak of the transmittance curve increases significantly as the film strain increases, and the corresponding included angle gradually shifts towards 90°. It can be concluded that the included angle designs of 0° and 90° are not equivalent when the substrates are under strain. Based on the case of this application example, the design of the included angle of 0° is more reasonable. Thus, prior knowledge of the anisotropy of the stress-optical properties of flexible films is the key to the lay-up of flexible films in flexible LCDs.

## 4. Conclusions

In this paper, the anisotropy of the stress-optical properties of PET films under a uniaxial stress state is quantitatively characterized. Transmission and reflection photoelastic experiments were carried out by using a reflection-transmission coaxial bidirectional photoelastic measurement system, and the stress-optical coefficients and isoclinic angles of PET films in different stretching directions were obtained. Both transmission and reflection photoelastic experimental results show that the birefringence of PET films remains linear in response to uniaxial stress for different stretching directions. Meanwhile, the strong anisotropy of the stress-optical properties of PET films is demonstrated. Specifically, the birefringence increases, the isochromatic phase decreases and the stress-optical coefficient is negative when the stretching direction is close to the slow axis direction. In contrast, the birefringence decreases, the isochromatic phase increases and the stress-optical coefficient is positive when the stretching direction is close to the fast axis direction. In addition, the analytical equation of the stress-optical coefficient versus isoclinic angle under the uniaxial stress state of the PET film was established, and the linear combinations of the photoelastic tensor components and refractive-index-related parameters were determined by fitting the experimental data. This result lays the foundation for precise control of the optical properties of PET films during deformation. We expect that the photoelastic method will be applied to more fundamental studies related to the stress-optical properties of polymer films, especially for the substrate materials of flexible electronic devices.

## Figures and Tables

**Figure 1 polymers-14-03257-f001:**
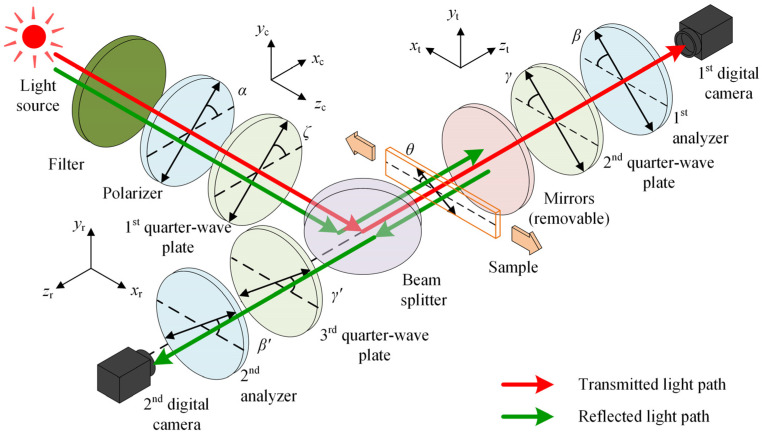
Schematic diagram of the photoelasticity measurement system.

**Figure 3 polymers-14-03257-f003:**
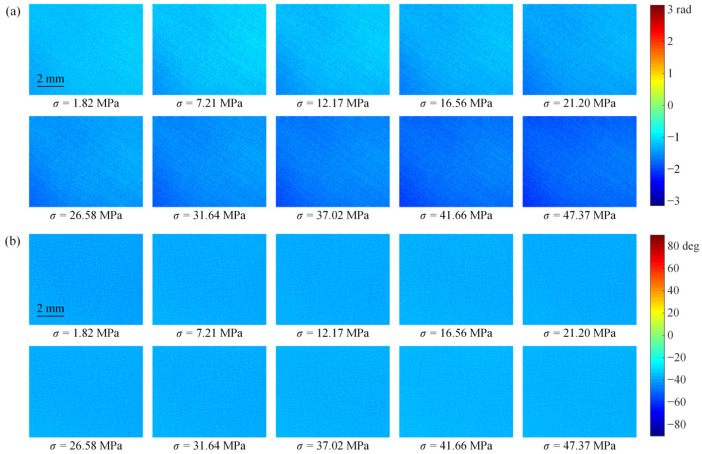
The full-field (**a**) isochromatic phase and (**b**) isoclinic angle distribution of the PET specimen with *φ* = −15° obtained in the transmission photoelastic experiment.

**Figure 4 polymers-14-03257-f004:**
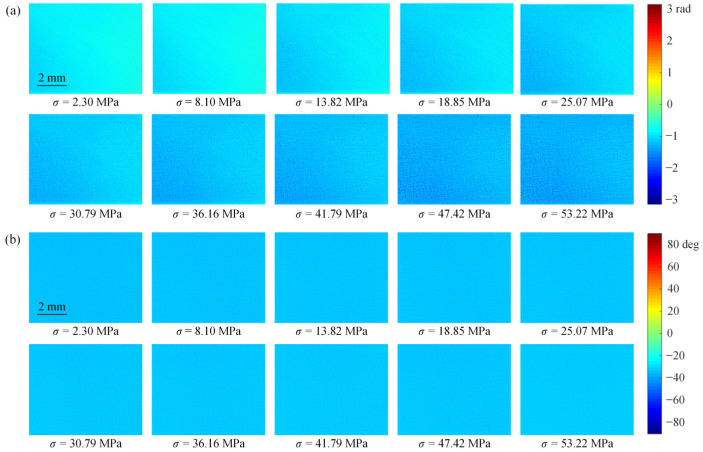
The full-field (**a**) isochromatic phase and (**b**) isoclinic angle distribution of the PET specimen with *φ* = −15° obtained in the reflection photoelastic experiment.

**Figure 5 polymers-14-03257-f005:**
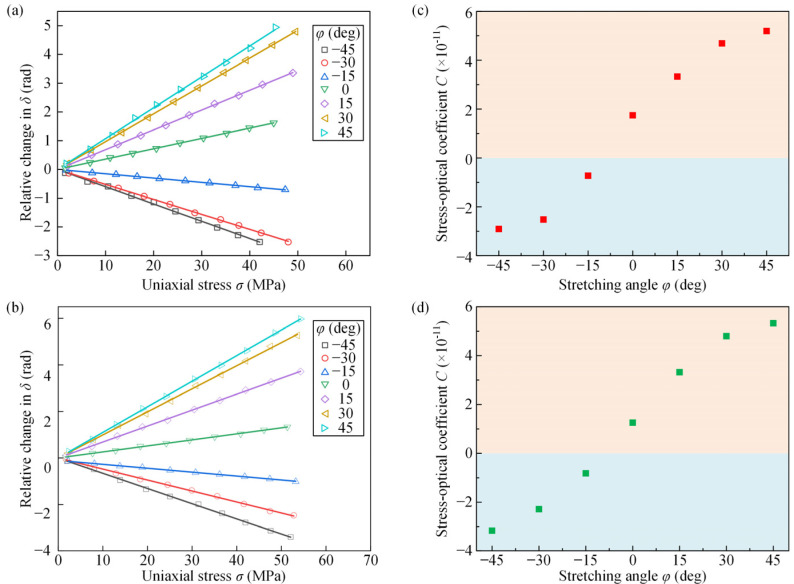
The relation between the relative change in the isochromatic phase and uniaxial stress of PET films with different stretching angles in (**a**) transmission photoelastic experiments and (**b**) reflection photoelastic experiments; (**c**,**d**) are the corresponding stress-optical coefficients.

**Figure 6 polymers-14-03257-f006:**
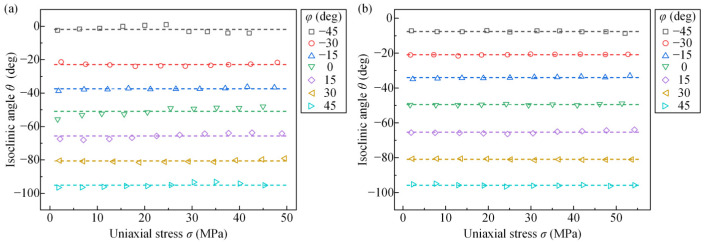
The relation between the isoclinic angle and uniaxial stress of PET films with different stretching angles in (**a**) transmission photoelastic experiments and (**b**) reflection photoelastic experiments.

**Figure 7 polymers-14-03257-f007:**
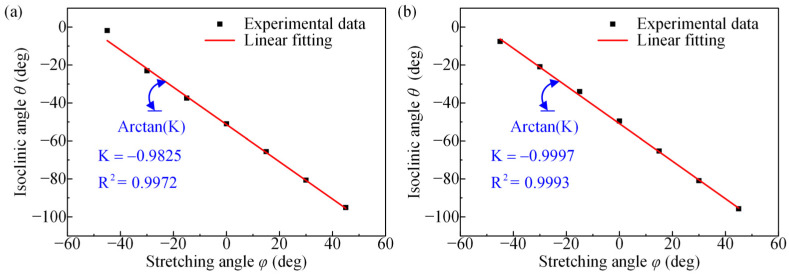
The relation between the isoclinic angle and stretching angles of PET films in (**a**) transmission photoelastic experiments and (**b**) reflection photoelastic experiments.

**Figure 8 polymers-14-03257-f008:**
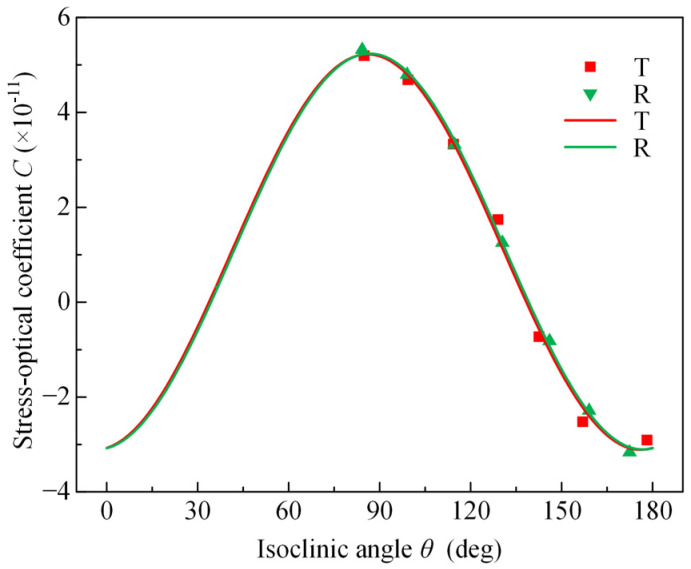
The relation between the isoclinic angle and the stress-optical coefficient of PET substrates. The fitting curve is compared with the experimental data.

**Figure 9 polymers-14-03257-f009:**
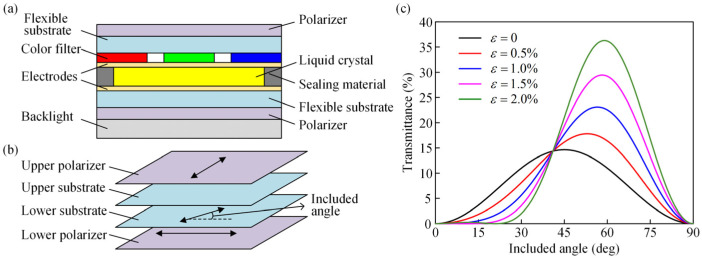
Guidance of flexible liquid crystal display applications by the anisotropy of the stress-optical properties. (**a**) Schematic diagram of the basic structure of a flexible liquid crystal display; (**b**) schematic diagram of the birefringence measurement of two flexible substrates; and (**c**) variation of the transmittance with the included angle under different strains.

**Table 1 polymers-14-03257-t001:** The light intensity equations in the plane polariscope.

Image Number	*α*	*β*	*β*′	Intensity Equation	
				T	R
1	π/2	0	0	$I_{1}=I_{b}+I_{a}\sin^{2}\frac{\delta }{2}\sin^{2}2\theta$	$J_{1}=J_{b}+J_{a}\sin^{2}\delta \sin^{2}2\theta$
2	π/4	0	0	$I_{2}=I_{b}+\frac{I_{a}}{2}(1-\sin4\theta \sin^{2}\frac{\delta }{2})$	$J_{2}=J_{b}+\frac{J_{a}}{2}(1+\sin4\theta \sin^{2}\delta )$
3	π/2	3π/4	π/4	$I_{3}=I_{b}+\frac{I_{a}}{2}(1+\sin4\theta \sin^{2}\frac{\delta }{2})$	$J_{3}=J_{b}+\frac{J_{a}}{2}(1-\sin4\theta \sin^{2}\delta )$
4	π/4	π/4	3π/4	$I_{4}=I_{b}+I_{a}\sin^{2}\frac{\delta }{2}\cos^{2}2\theta$	$J_{4}=J_{b}+J_{a}\sin^{2}\delta \cos^{2}2\theta$

**Table 2 polymers-14-03257-t002:** The intensity equations in the circular polariscope.

Image Number	*ζ*	*γ* = *γ*′	*β* = *β*′	Intensity Equation	
				T	R
5	3π/4	π/4	π/2	$I_{5}=I_{b}+\frac{I_{a}}{2}(1-\cos\delta )$	$J_{5}=J_{b}+\frac{J_{a}}{2}(1+\cos2\delta )$
6	3π/4	π/4	0	$I_{6}=I_{b}+\frac{I_{a}}{2}(1+\cos\delta )$	$J_{6}=J_{b}+\frac{J_{a}}{2}(1-\cos2\delta )$
7	3π/4	0	0	$I_{7}=I_{b}+\frac{I_{a}}{2}(1-\sin\delta \sin2\theta )$	$J_{7}=J_{b}+\frac{J_{a}}{2}(1+\sin2\delta \sin2\theta )$
8	3π/4	π/4	π/4	$I_{8}=I_{b}+\frac{I_{a}}{2}(1-\sin\delta \cos2\theta )$	$J_{8}=J_{b}+\frac{J_{a}}{2}(1-\sin2\delta \cos2\theta )$
9	π/4	0	0	$I_{9}=I_{b}+\frac{I_{a}}{2}(1+\sin\delta \sin2\theta )$	$J_{9}=J_{b}+\frac{J_{a}}{2}(1-\sin2\delta \sin2\theta )$
10	π/4	3π/4	π/4	$I_{10}=I_{b}+\frac{I_{a}}{2}(1+\sin\delta \cos2\theta )$	$J_{10}=J_{b}+\frac{J_{a}}{2}(1+\sin2\delta \cos2\theta )$

**Table 3 polymers-14-03257-t003:** Fitting results of the linear combinations of the photoelastic tensor components and refractive-index-related parameters.

	$\frac{n_{2}^{2}n_{1}^{2}}{n_{2}+n_{1}}$	$p_{11}-p_{12}({\mathrm{TPa}}^{-1})$	$p_{12}-p_{22}({\mathrm{TPa}}^{-1})$	$p_{16}-p_{26}({\mathrm{TPa}}^{-1})$	$R^{2}$
T	2.250	23.06	−13.66	5.092	0.9954
R	2.250	23.17	−13.69	4.158	0.9993

## Data Availability

The data used to support the findings of this study are included within the article.

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
