# Peer review of "Quantitative Characterization of the Anisotropy of the Stress-Optical Properties of Polyethylene Terephthalate Films Based on the Photoelastic Method"

_polymers, 2022, doi:10.3390/polym14163257_

Round 1

Reviewer 1 Report

Summary: in this manuscript, the authors established a photoelastic method to test the stress-optical properties of PET film. The method and the modeling are clearly explained. The major issue is why the retardation values are different at stress = 0 for different stress angles in Figure 5 (a) and Figure 5 (b). This inconsistency will impact the statement in line 316 to line 329.

Major issue:

(1) in Figure 5 (a) and Figure 5 (b), the retardation value at stress = 0 are different for different stress angles, which doesn’t make sense. When stress = 0, retardation should be equal to the original retardation no matter what stress angle is. In addition, the authors should provide the measured retardation values as reference.

(2) based on the authors’ definition (eq 10), retardation δ is defined as below

Where  and  are the refractive indices of the fast and slow axes of the specimen after stress is applied (line 225). Therefore,  <  and retardation is negative before stress is applied and after stress is applied. However, in Figure 5(a) and Figure 5(b), positive retardation values are presented at stress = 0 or when stress is applied. This is confusing. I think the authors need to check their definition of the retardation.

The original retardation and the definition of retardation (whether it should be negative or positive) can significantly impact the statement in line 316 to 329. For example, the authors claimed the original fast axis angle of the film (stress = 0) ~ -52o. In theory, when stress is applied along fast axis (close to -52o, in this work it is -45o and 30o), the absolute retardation value should decrease first (at small stress) because refractive index along fast axis increases more than slow axis, but Figure 5b shows a different trend for stress angle = -45o and 30o.

Overall, the authors should check (1) why the retardation values are different at stress = 0 for different stress angles; (b) check the definition of the retardation and why the definition is inconsistent with the values (especially positive and negative) in figure 5.

Reviewer 2 Report

The paper presents in detail a well done experiment, aimed at the characterization of the photoelastic tensor of the material. However there are some point that should be clarified.

i) the authors speak of anisotropy  but didn't mention the symmetry group of the  material: moreover they speak of "elastic modulus" but they didn't mention to which direction this modulus is referred. By looking  at equation (6) one may argue that the material is monoclinic: however in this case the matrix should not be symmetric. Please state clearly which symmetry group we are dealing with.

(ii) The reader has the wrong feeling that the authors aim to a complete characterization of the photoelastic matrix, whereas they obtain only a linear combination of the coefficients $p_{11}, p_{22}, p_{12}, p_{16}, p_{26}$. To this regard, since the matrix (7) should be not symmetric if the material is monoclinic, the result could be wrong. Please fix these points.

Round 2

Reviewer 1 Report

Thanks to the authors to address my questions. But the authors' explanation in Line 333 to Line 338 still doesn't make sense to me. As the authors explained, when the stress is along fast axis, birefringence increases because of due to refractive index along fast axis decreases more; and when stress is along slow axis (such as 45 degree), refractive index along slow axis decrease more.  Firstly, it is rare the stress along one direction (either slow axis or fast axis) cause refractive index decreases. I think the authors should check refractive index with other tools to confirm the conclusions; secondly, the authors statement is not even supported by their data: when the stress is close to slow axis (for example 45 deg), the authors said the birefringence decreases, which is not true; as shown in Figure 5(a) and Figure 5 (b), the birefringence gets more and more negative with higher stress, which means absolute birefringence value increases with higher stress. I think the authors need to double check their statements.

Round 3

Reviewer 1 Report

I appreciate the authors to validate the refractive index and modify the manuscripts based on the previous comments and the new experiment results.